# Unveiling the Potential of Quantization with MXFP4: Strategies for Quantization Error Reduction

**Jatin Chhugani** [* 1]   **Geonhwa Jeong** [* 1]   **Bor-Yiing Su** [1]   **Yunjie Pan** [1]   **Hanmei Yang** [1]   **Aayush Ankit** [1]   **Jiecao Yu** [1]   **Summer Deng** [1]   **Yunqing Chen** [1]   **Nadathur Satish** [1]   **Changkyu Kim** [1]

## Abstract

Large Language Models (LLMs) have intensified the need for low-precision formats for efficient inference. The Open Compute Project Microscaling (MX) standard is attractive due to its favorable hardware efficiency, but its 4-bit variant (MXFP4) lags behind NVIDIA's NVFP4 in accuracy, limiting adoption. We introduce two software-only techniques, *Overflow-Aware Scaling* (OAS) and *Macro Block Scaling* (MBS), that improve MXFP4 quantization fidelity without requiring hardware changes. OAS reduces overall errors by increasing effective dynamic range under power-of-two block scaling, while MBS allocates higher-precision scaling at a coarser granularity to better preserve outliers. Across multiple LLMs and standard downstream benchmarks, OAS and MBS reduce the end-to-end accuracy gap between MXFP4 and NVFP4 from about 10% to below 1% on average, while incurring modest GEMM overhead (6.2% on average). These results re-establish MXFP4 as a practical alternative to NVFP4, enabling near-NVFP4 accuracy while retaining MX's hardware-efficiency advantages (e.g., 12% relative area savings in tensor cores).

## 1. Introduction

Large Language Models (LLMs) are rapidly transforming the landscape of artificial intelligence, driving breakthroughs across a wide range of applications. As the demand for higher performance continues to grow, researchers are scaling these models to unprecedented sizes. However, this scaling comes with significant computational and resource challenges, making efficiency a critical concern.

Quantization has emerged as a promising solution to address these challenges, enabling more efficient deployment of LLMs by reducing the precision of model parameters. Among various quantization formats, the microscaling format (MX) has gained traction and is becoming a standard, largely due to its adoption and promotion by multiple companies through the Open Compute Project (OCP) (Rouhani et al., 2023a). The MX proposal includes a family of formats, ranging from 8-bit/6-bit down to 4-bit formats. While there have been successful demonstrations of the 8-bit and 6-bit formats (Rouhani et al., 2023b; Mishra et al., 2025), preserving model quality with the MXFP4 format remains a significant challenge (Egiazarian et al., 2025; NVIDIA et al., 2025; Castro et al., 2026).

NVIDIA has proposed a new 4-bit format, NVFP4 (Alvarez et al., 2025) which has higher representation fidelity than the MXFP4 format. Recent studies also show NVFP4 preserves the model quality better (NVIDIA et al., 2025; Chen et al., 2025b; Egiazarian et al., 2025; Chen et al., 2025c; Chmiel et al., 2025). This fidelity gap poses a significant barrier to the widespread adoption of MXFP4 in scenarios where model quality is paramount. However, supporting the NVFP4 format incurs extra area and energy overheads to the hardware design. In this work, we perform a detailed analysis comparing the MXFP4 and NVFP4 in terms of representation fidelity and hardware costs. Building on these insights, we propose strategies to push the limits of MXFP4 quantization, achieving improved accuracy without requiring any hardware changes[1]. Our primary contributions are:

1. We identify the two primary sources of MXFP4's accuracy gap relative to NVFP4, coarser block granularity and power-of-two scaling precision, and quantify their fidelity and hardware-area trade-offs.

2. We propose *Overflow-Aware Scaling* (OAS) and *Macro Block Scaling* (MBS), two SW techniques that improve MXFP4 representation fidelity without requiring hardware modifications, making them applicable to MXFP4-compatible devices.

---

*Equal contribution [1]Meta Platforms, Inc., Menlo Park, California, USA. Correspondence to: Jatin Chhugani <jatinch@meta.com>, Geonhwa Jeong <geonhwa@meta.com>.

*Proceedings of the $43^{rd}$ International Conference on Machine Learning*, Seoul, South Korea. PMLR 306, 2026. Copyright 2026 by the author(s).

---

[1]While this work focuses on MXFP4, our proposed methods are generalizable to other MX formats, such as MXFP6 and MXFP8.

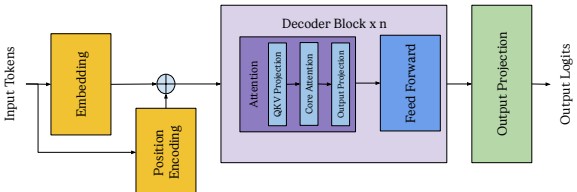

Figure 1. Modern LLM Model Architecture.

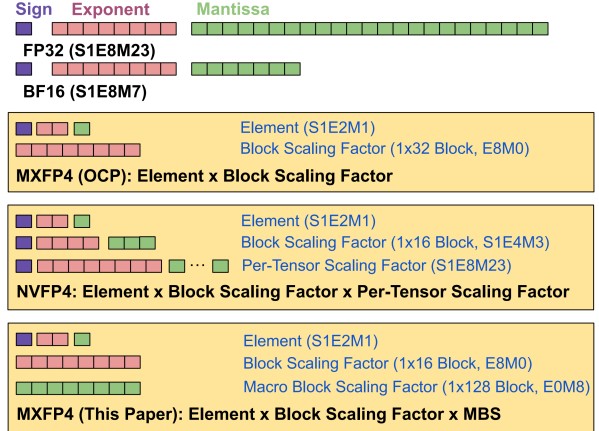

Figure 2. Comparison of different FP4 formats for quantization.

3. We demonstrate that enhanced MXFP4 achieves near-NVFP4 fidelity (within 1 dB QSNR) and downstream accuracy (within 1% on average), with modest GEMM overhead (6.2% on average), thereby unlocking MXFP4's hardware-efficiency benefits.

**Conflict of Interest Disclosure.** All authors are employees of Meta Platforms, Inc. The research was conducted utilizing publicly available, open-source models and benchmarks, and the authors declare no other competing interests.

## 2. Background

### 2.1. Transformer

Modern LLMs, including Llama 3 (Grattafiori et al., 2024), Llama 4 (Meta AI, 2025), Qwen (Team, 2025), DeepSeek (DeepSeek-AI et al., 2025), and GPT-OSS (OpenAI et al., 2025), are built upon the transformer architecture. Figure 1 illustrates the decoder-only transformer used by these models. The dominant computation arises from linear layers in the QKV projections, output projection, and feed-forward network (FFN). In this work, we focus on improving the efficiency of these linear layers through weight and activation quantization.

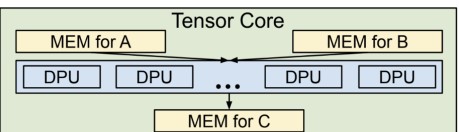

Figure 3. Hardware architecture of Tensor Core.

### 2.2. Quantization with FP4

Quantization reduces the precision of weights and activations to improve the efficiency of LLM inference. Common schemes vary in granularity (e.g., per-tensor, per-channel, or per-group) and scaling strategy (symmetric or asymmetric) (Su et al., 2025; Micikevicius et al., 2022; Xiao et al., 2024; Or et al., 2025). Furthermore, quantization-aware training (QAT) or post-training quantization (PTQ) techniques can be further employed to optimize model performance under reduced precision. Recent low-bit formats, such as MXFP4 and NVFP4, further explore this trade-off by enabling aggressive precision reduction while aiming to preserve model accuracy. These two formats represent prominent 4-bit quantization approaches for LLM deployment (Rouhani et al., 2023a; Alvarez et al., 2025). NVFP4, developed by NVIDIA, is widely adopted due to its strong accuracy and compatibility with existing hardware, whereas MXFP4, standardized by the Open Compute Project (OCP), is gaining attention for its improved HW efficiency. The key differences between these formats lie in their numerical encoding schemes and hardware implementation requirements, as shown in Figure 2.

Standard floating-point (FP) numbers comprise one sign bit, $E$ bits for the exponent, and $M$ bits for the mantissa. The OCP MXFP4 format comprises two components: 4-bit data elements in E2M1 and a shared E8M0 block scale applied to every 32 elements. In contrast, NVFP4 is composed of three components: the 4-bit data elements in E2M1, a shared E4M3 FP8 block scale applied to every 16 elements, and a per-tensor scaling factor to mitigate range limitations. While NVFP4 generally provides higher representation fidelity, MXFP4 offers substantial resource savings, making it attractive for large-scale and energy-efficient deployments.

Our proposed advanced MXFP4 format is similarly defined as a combination of three components to enhance fidelity. The details of this format are provided in Section 4.3.

### 2.3. HW for MXFP4 GEMM and NVFP4 GEMM

General Matrix Multiplication (GEMM) operations are central to LLM inference. Hardware support for GEMM using MXFP4 and NVFP4 formats varies based on the underlying architecture. Figure 3 shows the hardware implementation of a typical tensor core (Zhu et al., 2019; Hickmann et al., 2020; Darvish Rouhani et al., 2023) which can support

MXFP4/NVFP4 as input data types. Memory for A and B store the input operand matrices and Memory for C stores the partial sums or final output. Multiple Dot Product Units (DPU) instantiated based on Performance, Power, Area constraints, perform a HW tile size for the target GEMM operation. We present our detailed comparison on HW overhead of the different FP4 formats in Section 3.4.

# 3. Understanding NVFP4 vs. MXFP4

## 3.1. Analysis Methodology

In this paper, we evaluate representational fidelity using the Quantization Signal-to-Noise Ratio (QSNR), measured in decibels (dB). While different metrics can be used, we adopt QSNR (Darvish Rouhani et al., 2023) as it exhibits strong correlation with end-to-end metrics of inference quality. QSNR is also used by other works (Darvish Rouhani et al., 2023; Egiazarian et al., 2025) to derive a first-order analysis of the different formats. We compute QSNR at two granularities: for individual (input) tensors and for post-operation results (e.g., the output of matrix multiplication). This allows us to normalize the Mean Squared Error (MSE) of the quantized tensors relative to their high-precision counterparts. A higher value of QSNR signifies a lower error due to quantization, and hence higher fidelity.

Formally, let $A^{\text{BF16}}$ denote the tensor in the original high precision and $A^Q$ denote the quantized tensor. The QSNR is defined as the logarithmic ratio of the reference signal power to the quantization noise:

$$\text{QSNR}(A^{\text{BF16}}, A^Q) = 10 \log_{10} \left( \frac{\|A^{\text{BF16}}\|_F^2}{\|A^{\text{BF16}} - A^Q\|_F^2} \right) \quad (1)$$

$$\text{QSNR}(AB) = 10 \log_{10} \left( \frac{\|A^{\text{BF16}} B^{\text{BF16}}\|_F^2}{\|A^{\text{BF16}} B^{\text{BF16}} - A^Q B^Q\|_F^2} \right) \quad (2)$$

In this paper, we conduct QSNR analysis on two popular LLMs, Llama 3.1-8B-Instruct and Qwen3-8B. We use the tensors dumped during the inference of each model. We randomly sampled 1000 tensors and use the average value for the analysis.

## 3.2. Implications of Fine-Grained Block Quantization: (32 → 16)

The limited exponent width (E = 2) of FP4 inherently constrains the representable dynamic range to a very small ratio: $12\times \, (= 6.0/0.5)$. Consequently, blocks exhibiting high variance inevitably incur increased flush-to-zero (values quantized to zero) ratio for smaller magnitude values. For example, for activation tensors, decreasing the block size from 32 to 16 results in a reduction in flush-to-zero values from 20% to 13%, i.e. decreasing the flush-to-zero

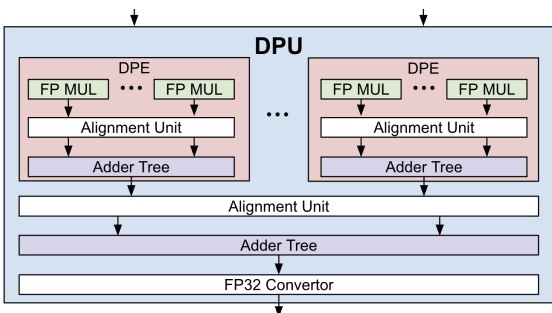

*Figure 4.* Overview of the DPU architecture.

ratio by 35%. While techniques such as using rotation matrices (Egiazarian et al., 2025) help reduce the dynamic range of the blocks, the relative decrease in flush-to-zero rates remains similar. Consequently, the reduced block size yields a net increase in QSNR around **1 dB**.

## 3.3. Impact of Fine-Grained Scaling Factor Format: E8M0 → E4M3

While the scale factor formats differ in exponent width (4-bit vs. 8-bit) between NVFP4 and MXFP4, the extended range of the 8-bit exponent in MXFP4 is largely redundant. Notably, for nearly all weight tensors and over 98% of activation tensors, a 4-bit exponent suffices to capture the scaling factor's dynamic range. Consequently, the MXFP4 scaling factor format leaves four exponent bits unutilized for most tensors—an inefficiency we propose addressing by truncating the exponent from E8 → E4 for compact storage when needed.

The critical functional distinction lies in the mantissa bits. Given that systematic tensor outliers govern quantization thresholds, their precise resolution is paramount for fidelity (Dettmers et al., 2022; Xiao et al., 2024). However, MXFP4 (E8M0) lacks mantissa bits, rigidly constraining scaling factors to powers-of-two. This prevents the accurate representation of outliers falling between intervals; for instance, values between $4.0$ and $6.0$ can incur representation errors of up to $20\%$. Conversely, the E4M3 scaling format (NVFP4) retains three mantissa bits, enabling finer-grained scaling precision that can better approximate the optimal scale for these critical outliers (within 0.2-0.3dB of storing an FP32 scaling factor). Thus, E4M3 effectively minimizes the error for large-magnitude values for the given budget of 8-bits, thereby significantly boosting the tensor's QSNR. We observed a **3–4 dB** improvement attributable solely to increased scaling factor mantissa precision and analyze the specific hardware costs of this fidelity gain next.

## 3.4. Impact on HW Cost of Block Size and Scaling Factor Format

To understand the HW cost, we extract the baseline area numbers of main components (element-wise matrix multi-

plication, adder trees, shifters and so on) from a production implementation of a multi-format tensor core similar to previous work (Zhu et al., 2019; Hickmann et al., 2020; Lee et al., 2022; Darvish Rouhani et al., 2023) mapped to an advanced TSMC tech node. We use these numbers to build an analytical area model to compare trade-offs between NVFP4 and MXFP4, specifically the impact of (i) the scale factor block size (32 vs. 16) and (ii) the scale factor format (E8M0 vs. E4M3). For a fair comparison, we assume identical hardware tile sizes across designs for tensor cores. Tensor core area can be attributed to two major components: memory and compute logic.

**Block Size Overhead.** We first measure the area impact of using a scale-factor block size of 16 instead of 32. To be conservative, we assume E8M0 scale factors. Based on our area model, a block size of 16 increases tensor-core area by 2% relative to a block size of 32. This increase is mainly attributed to slightly higher size of SRAM (4.5 bits per element for a block size of 16 compared to 4.25 bits per element for 32) needed for A/B memories and an increase in inter-block adder tree width.

**Scale Factor Format Overhead (E4M3 vs. E8M0).** Next, fixing the block size to 16, we quantify the area overhead of using E4M3 rather than E8M0 as the scale-factor format. Memory capacities for A and B are identical in both cases because the block size is fixed. For example, with block size 16, an 8-bit scale factor, and 4-bit data, each block requires $16 \times 4 + 8 = 72$ bits. The capacity of memory for C is also unchanged because the output precision (assumed to be FP32) does not depend on the input format. Therefore, E8M0 and E4M3 do not differ in memory capacity under the same block size.

In terms of compute logic (i.e., DPU in Figure 4), the main difference arises in the inter-block alignment logic across DPEs (Dot Product Engines), which (1) resolves the effective scale for each block, (2) computes the maximum exponent ($E_{\max}$), and (3) aligns block-level mantissas with respect to $E_{\max}$ while adding the partial sums across blocks.

- **With floating-point scale factors (E4M3):** Scale resolution requires floating-point multiplication. The shared scale is applied to values within the block, requiring TensorCoreTileSize/BlockSize floating-point multiplications. Moreover, the maximum exponent must be determined across all elements (i.e., NumBlocks × BlockSize values).

- **With power-of-two scale factors (E8M0):** Scale resolution requires only one integer addition per block. The shared scale exponent is added to the block maximum exponent to obtain the effective exponent for the block. The global maximum exponent is then computed by comparing only one value per block.

*Table 1.* Relative Tensor Core Area (MXFP4 / NVFP4) across varying HW configurations.

| Logic Fraction (Multiplier=0.2) | Relative Area | Multiplier Fraction (Logic=0.5) | Relative Area |
|---|---|---|---|
| 0.3 | 0.93 | 0.10 | 0.94 |
| 0.4 | 0.91 | 0.15 | 0.91 |
| 0.5 | 0.88 | 0.20 | 0.88 |
| 0.6 | 0.86 | 0.25 | 0.86 |
| 0.7 | 0.84 | 0.30 | 0.83 |

As a result, inter-block alignment is substantially more expensive for E4M3. Based on our proprietary area model, using E4M3 as the scale-factor format incurs a 21.3% compute logic area overhead, translating to a 12.6% total tensor core area overhead relative to E8M0.

**Parametric Area Analysis.** Tensor Core (TC) configurations vary across generations and vendors based on dimensions, spatio-temporal tiling, and proprietary design choices. Because our absolute area numbers stem from a proprietary EDA flow, we abstract the key parameters into two ratios: the *logic fraction* (fraction of TC area occupied by compute datapath) and the *multiplier fraction* (fraction of compute datapath occupied by element-wise multipliers). As storage area is identical between formats, a TC configuration that is more compute-dominated sees larger MXFP4 savings. Furthermore, because NVFP4 requires expensive FP multipliers for inter-block alignment, a TC configuration with more expensive multiplier implementations adds to the NVFP4 overhead. As shown in Table 1, across all reasonable parameter choices, MXFP4 TC consistently shows meaningful area reduction (7% to 17%) over NVFP4 TC.

**Cross-Validation with a Public Area Model.** To ensure reproducibility, we cross-validate our proprietary estimations using a publicly available analytical HW cost model (Chen et al., 2025a). Their model decomposes a Matrix-Multiply Unit (MMU) into three components: MAC (multiply-accumulate), DEQ (dequantizer / inter-block alignment), and ACC32 (FP32 accumulator), counting standard cells (FA, HA, MUX, AND gates) based on a TSMC FinFET standard-cell library.

Using their methodology with standard multi-bit adder and multiplier gate counts, we derived the comparison between MXFP4 and NVFP4 tensor cores at block size 16 as shown in Table 2. The MAC and accumulator costs are identical between formats, and the entire overhead stems from the DEQ (inter-block alignment), where NVFP4 requires floating-point multiplications to resolve the E4M3 scales compared to simple integer additions for MXFP4's E8M0 scales. Due to this large DEQ gap, NVFP4 incurs a significant area overhead, which is consistent with the analysis using our area model.

*Table 2.* Gate count ratio comparison of Matrix-Multiply Unit (MMU) sub-components. Each number is normalized to the gate counts of MAC of MXFP4.

| Component | MXFP4 | NVFP4 |
|---|---|---|
| MAC | 1.000 | 1.000 |
| DEQ | 0.026 | 0.262 |
| ACC32 | 0.311 | 0.311 |
| **Total** | **1.337** | **1.573** |

## 3.5. Proposed Direction

In summary, we attribute the fidelity gap to fundamental granularity differences between NVFP4 and MXFP4 across two axes: (1) block size and (2) scale factor format. While both improve fidelity, our analysis reveals a clear trade-off: fine-grained scale factors incur high hardware costs, whereas reducing block size is inexpensive.

Consequently, we adopt a finer block size of 16 to leverage spatial locality while retaining the cost-effective coarser E8M0 scale factor format. To recapture the precision of a fine-grained scale factor format without the associated hardware cost, we propose *Overflow-Aware Scaling (OAS)* along with *Macro Block Scaling (MBS)*. This approach allows area-efficient MX hardware to achieve fidelity competitive with NVFP4, effectively decoupling high model performance from expensive hardware requirements.

## 4. Enhancing the MX Format

### 4.1. Quantization Block Granularity

As detailed in Section 3.2, reducing the block size is imperative for low-precision formats like FP4. While the standard OCP specification defines MXFP4 with a default block size of 32, NVFP4 utilizes a finer block size of 16 (NVIDIA Corporation, 2025a). Furthermore, we found out that NVIDIA's Blackwell architecture natively supports MXFP4 GEMM operations at this 16-element block size. This minimal adjustment to reduce block size from 32 to 16 recovers 1 dB QSNR. We have added the native MXFP4 GEMM with 16-element block size in the MSLK library (Meta PyTorch, 2026).

### 4.2. Overflow-Aware Scaling (OAS)

Following standard quantization routines, for each $1 \times 16$ block, we compute $\text{SF}_{\text{FP32}} = 6.0/\alpha_{\max}$ (given FP4 $\text{FP}_{\max} = 6.0$) where $\alpha_{\max}$ is the absolute maximum value in the block. We obtain the E8M0 scale by masking mantissa bits to enforce the power-of-two constraint. The standard computation ensures that $\alpha_{\max}$ maps to the representable range, $(3, 6]$, preventing saturation (clamping) error. When $\alpha_{\max} \in (3, 3.5]$ (3.5 being the mid-point of the two con-

secutive representable values from the half of the largest representable value, i.e. 3 and 4), doubling the scaling factor maps the $\alpha_{\max}$ to $(6, 7]$ results in saturation since the format limit (i.e. the largest representable value) is 6.0. Nevertheless, we observe that this shift **preserves the relative quantization error** for $\alpha_{\max}$ (e.g., quantizing $3.3 \mapsto 3.0$ versus $6.6 \mapsto 6.0$ yields identical relative error). More broadly, this scaling adjustment maintains relative error fidelity for any block element that previously mapped to the standard FP4 normal range of $[1, 6]$. In addition, the key advantage of this approach is that **it doubles the representable dynamic range** to accommodate lower-magnitude elements, thereby reducing quantization error for the tail of the distribution. We call this Overflow-Aware Scaling (OAS), which applies scaling to map $\alpha_{\max}$ to $(3.5, 7]$. It is also worthwhile to note that MXFP4-OCP maps $\alpha_{\max}$ to $[4, 8]$, which allows some overflow, but not ideal unlike OAS. For example, if $\alpha_{\max}$ is mapped to 7.6, the quantization error would be $|\frac{(7.6-6)}{7.6}| = 21\%$, but if it was mapped to 3.8, then the error could have been $|\frac{(3.8-4)}{3.8}| = 5.3\%$. We also analyzed 100 dumped tensors from Llama-3.1-8B-Instruct inference; this includes 23.4 million $1 \times 16$ blocks. If $\alpha_{\max}$ is mapped in $(7, 8)$, the corresponding blocks are the ones where OAS triggers, shifting them to $(3.5, 4)$. We observe OAS is triggered for 18.03% of all blocks, shifting scaled $\alpha_{\max}$ from $(7, 8)$ to $(3.5, 4)$ as they incur larger quantization error. We also found a similar trend on dumped tensors from Qwen3-8B: 18.64% of the blocks were triggered based on the OAS threshold compared to OCP-style scaling.

We implement OAS by checking mantissa bits and the pseudocode is summarized in Appendix A. We observe around 0.5 dB increase in terms of QSNR by using OAS compared to MXFP4-OCP without adding additional overhead. It also largely improves downstream evaluation results as shown in Section 5.

### 4.3. Macro Block Scaling (MBS)

Outliers play a disproportionate role in quantization fidelity, despite comprising a negligible fraction (typically less than 1%) of each tensor (Guo et al., 2023; Dettmers et al., 2023). A fundamental limitation of the E8M0 scaling format is that its quantization error is strictly a function of the original value, meaning the format lacks the flexibility to prioritize or "attend" to these critical outlier regions, irrespective of the scaling factor used as it does not change mantissa bits.

To address this, we propose coarser scaling (specifically targeting a $1 \times 128$ block size) with higher precision (with 8 bits of mantissa) as the **Macro Block Scaling (MBS)**. Although this granularity is coarser than the fundamental compute block size of $1 \times 16$, we identify $1 \times 128$ as the optimal compromise as shown in Appendix B: it is sufficiently fine-grained to isolate high-magnitude outliers effectively

with extra mantissa bits, yet coarse enough to minimize post-processing overhead and storage costs[2]. It is important to note that while both NVFP4 and our proposed MBS approach differ in the approach to store/process scaling factors, our method achieves a critical advantage: it effectively isolates outliers with MBS to preserve model fidelity without the prohibitive hardware cost associated with native fine-grained scaling format (i.e. E4M3 for local scale factor). In addition, our quantization strategy maintains minimal computational overhead by eliminating the need for a two-pass traversal over the tensor for scale computation and subsequent quantization. Our MBS scheme operates on local $1 \times 128$ blocks, which constitutes a natural extension of the fundamental $1 \times 16$ granularity and is handled seamlessly within the existing CUDA parallelization framework.

### 4.3.1. COMPUTATION OF MBS FACTOR

We define the macro-block maximum as $\alpha_{\max}^{128} = \max(\alpha_1^{16}, \ldots, \alpha_8^{16})$, where $\alpha_i^{16}$ denotes the maximum of the absolute values (absmax) within the $i$-th contiguous $1 \times 16$ sub-block. We assume an algorithm that maps the input $\alpha_{\max}^{128}$ to a scale factor $\mathrm{SF}_{\mathrm{MBS}}^{128}$ (e.g., $\mathrm{SF}_{\mathrm{MBS}}^{128} = 6.0/\alpha_{\max}^{128}$). We express the scaling factor as $\mathrm{SF}_{\mathrm{MBS}}^{128} = 2^e(1 + m_{\mathrm{MBS}})$ similar to GAM (Su et al., 2025). Since the local $1 \times 16$ E8M0 scales are purely exponential, they efficiently subsume the macro-exponent $e$, requiring storage only for the mantissa. Empirically, an 8-bit representation approximates the scale within $0.3\%$. Consequently, we propose storing only the quantized mantissa, denoted as $m_{\mathrm{MBS}}^8$. Eventually, we use $(1 + m_{\mathrm{MBS}}^8)$ as the actual MBS Factor so the $1 \leq \mathrm{MBS}\ \mathrm{Factor} < 2$. Upon computing $(1 + m_{\mathrm{MBS}}^8)$, we scale the elements of each $1 \times 16$ block by this factor. This operation shifts the input distribution into the optimal range before we apply the standard MXFP4 quantization, while leveraging our OAS. The MBS computation pseudocode is available in Appendix A.

### 4.3.2. MATRIX MULTIPLICATION WITH MBS

We perform the matrix multiplication $AB^T$, where $A \in \mathbb{R}^{M \times K}$ and $B \in \mathbb{R}^{N \times K}$. The computation adheres to a tiled execution model (e.g., CUTLASS (NVIDIA Corporation, 2025a)), wherein discrete tiles of both matrices are iteratively fetched from HBM into the cache hierarchy. We define the tile dimensions as $T_M \times T_K$ for matrix $A$ and $T_N \times T_K$ for matrix $B$. Consequently, the inner kernel executes the product of these tiles, denoted as $(T_M \times T_K) \times (T_N \times T_K)^T$.

From a theoretical perspective, we can align the kernel

---

[2]Spatially clustered outliers could potentially benefit from pre-processing techniques such as column reordering (Zhao et al., 2024); however, integration of such permutation-based optimizations remains beyond the scope of this work.

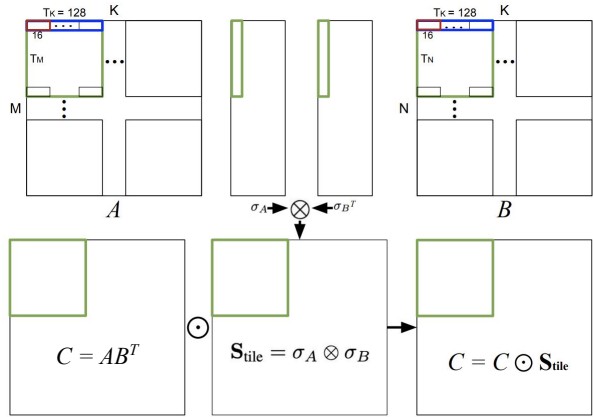

*Figure 5.* Matrix multiplication $AB^T$ with MBS.

to $1 \times 128$ MBS granularity ($T_K = 128$). While the native `tcgen05.mma.kind::mxf4` instruction family processes 64-element chunks (NVIDIA Corporation, 2025c) in the K dimension, we can intercept the GEMM at $128^3$ tile granularity. This synchronization (Figure 5) ensures scaling updates coincide with tile boundaries, enabling efficient epilogue interception without architectural divergence.

At initialization, threads prefetch encoded MBS ($m_{\mathrm{MBS}}^8$) to LLC and compute FP16 scales $\sigma = (1 + m_{\mathrm{MBS}}^8)^{-1}$. The steady-state loop employs a multi-stage pipeline to hide latency, issuing asynchronous instructions (e.g., `cp.async`) to stage subsequent tiles while concurrently saturating Tensor Cores with the compute-bound FP4 GEMM. Upon loop termination, the $128 \times 128$ FP32 output tile $\mathbf{C}_{\mathrm{tile}}$ resides in the distributed register file. In the epilogue, we synthesize the de-quantization surface $\mathbf{S}_{\mathrm{tile}} = \sigma_A \otimes \sigma_B$ and fuse it directly into the accumulators via an element-wise Hadamard product: $\mathbf{C}_{ij} \leftarrow \mathbf{C}_{ij} \odot (\sigma_{A,i} \cdot \sigma_{B,j})$. This operation maps to a sequence of low-latency, register-level FP32 FMUL instructions prior to writeback.

From a Roofline perspective (Appendix C), MBS latency is theoretically hidden provided Vector Core throughput exceeds $\approx 1.56\%$ (1/64) of Tensor Core peak. We quantify realized projected overhead in Section 5.3. Crucially, we schedule MBS scaling on Vector Cores concurrently with the main workload, leaving Tensor Cores fully dedicated to the dense GEMM. Thus, MBS can be seen as a software optimization, requiring no hardware changes.

### 4.3.3. OPTIMIZATION FOR MBS

For the MBS scheme, we compute the scaling factor $(1 + m_{\mathrm{MBS}}^8)$ via two proposed algorithms, differentiated by their trade-off between computational overhead and fidelity (QSNR and end-to-end accuracy).

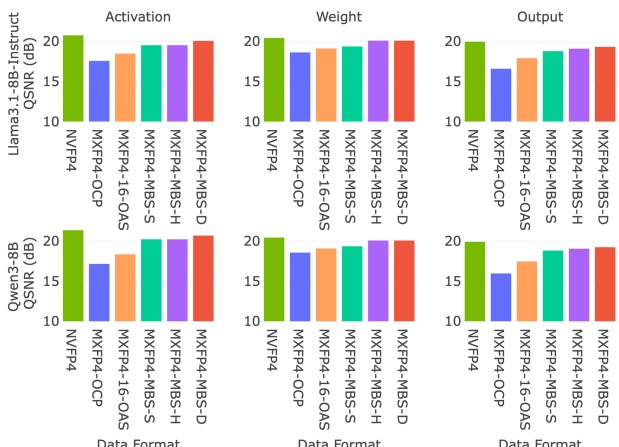

*Figure 6.* QSNR analysis with different formats on activation, weight, and output of matrix multiplication.

**Static:** We derive the scaling factor from the macro-block maximum $\alpha_{\max}^{128}$, directly computing the reciprocal to normalize to $F_{\max} = 6.0$ and extracting the 8 most significant bits:

$$m_{\text{MBS}}^8 = \left( \text{bits}\left( \frac{6.0}{\alpha_{\max}^{128}} \right) \,\&\, \texttt{0x007F8000} \right) \gg 15 \quad (3)$$

This operation isolates the target scale's leading mantissa bits, yielding a computationally inexpensive approximation. In terms of fidelity, MBS-Static (**MBS-S**) improves average QSNR by $+1.1$ dB over the MXFP4 using $1 \times 16$ with OAS.

**Dynamic:** While the Static assignment of $m_{\text{MBS}}^8$ is robust, it lacks MSE guarantees (Egiazarian et al., 2025). We address this via a memoization-based search over a narrow range, trading marginal overhead for superior fidelity.

**Memoization Strategy.** To bypass runtime MSE calculation, we optimize MBS selection via a precomputed Look-Up Table (LUT). For a candidate factor $m_j$ and input $x_i$, we conceptually scale to $x_i \cdot (1 + m_j)$, derive the local scaling factor $SF$ with OAS from the block's maximum magnitude, and define the quantized output as:

$$\hat{x}_i = Q^{\text{FP4}}\left( x_i \cdot (1 + m_j) \cdot SF \right) \quad (4)$$

These tables store squared relative errors, indexed by candidate scale $m_j$ and scaled intermediate value $v_{ij} = x_i \cdot SF$.

$$\mathcal{T}[v_{ij}, m_j] \approx \left( \frac{\hat{x}_i - x_i}{x_i} \right)^2 \quad (5)$$

We route sub-normal ($v_{ij} < 1$) and normal ($v_{ij} \geq 1$) values to distinct tables, discretizing the domain into 64 points with 16 slots. The resulting 2,048 entries (4KB in FP16) occupy $< 2\%$ of NVIDIA B200 shared memory. Runtime access

ensures fully coalesced shared memory reads, with the final factor $m_{\text{MBS}}^8 = m_{j*}$ selected by minimizing macro-block Sum of Squared Errors (SSE):

$$j^* = \arg\min_j \sum_{b=1}^{8} \sum_{i=1}^{16} (x_{bi})^2 \cdot \mathcal{T}[v_{bi,j}, m_j] \quad (6)$$

*Overhead Analysis:* The search entails one type conversion and one FFMA per slot, costing $\sim 32$ ops/element amortized. This fixed per-element cost is negligible relative to the GEMM workload, as it is diluted by the massive, $K$-scaling arithmetic intensity of the core kernel.

*Fidelity Improvement:* **MBS-D** yields a 1.6 dB QSNR improvement over the MXFP4 using $1 \times 16$ with OAS. We also show that these QSNR gains strongly correlate with the recovery of downstream model accuracy (Section 5) consistent with the previous work (Darvish Rouhani et al., 2023; Egiazarian et al., 2025).

### 4.4. Overall QSNR Comparison of NVFP4 with MX4-MBS-[S/D]

As shown in Figure 6, MBS elevates QSNR from $18.6 \rightarrow 20.1$ dB (Weights) and $17.4 \rightarrow 19.9$ dB (Activations), narrowing the NVFP4 gap to $< 1$ dB. Given the strong positive correlation between QSNR and end-to-end model accuracy, this proximity implies *statistically similar errors* and comparable inference quality. Balancing fidelity with runtime cost, we employ MBS-Dynamic for Weights and MBS-Static for Activations, designating this configuration **MBS-Hybrid (MBS-H)** as our default for all end-to-end evaluations.

## 5. Evaluation

### 5.1. Setups

To evaluate the proposed enhanced MX formats, we use vLLM (Kwon et al., 2023) as the inference engine and use Language Model Evaluation Harness (Gao et al., 2024) as the evaluation framework. We use Llama 3.1-8B (Grattafiori et al., 2024), Qwen3-8B (Team, 2025), Llama 4-Maverick (Meta AI, 2025), and DeepSeek-R1 (DeepSeek-AI et al., 2025). We only quantize linear layers (including QKVO projections, linear layers in FFN, and each expert in MoE layers). We apply fake quantization to both weight and activation for all our end-to-end evaluation benchmarks. The fake quantization is implemented by scaling the block with the scaling factor, applying quantization to the target format, dequantizing to BF16, and then un-scaling back to the original numerical range. To focus on the effectiveness of the format, we use direct-cast without using any calibration data following the methods used in the previous works (Darvish Rouhani et al., 2023; Lee et al., 2025), so our evaluation **does not** use any re-training and fine-tuning.

*Table 3.* Downstream evaluation results on **Llama3.1-8B-Instruct** with different formats.

| PRECISION | MMLU-PRO | GSM8K | HELLASWAG | WINOGRANDE | ARC-C | ARC-E | **AVERAGE** |
|---|---|---|---|---|---|---|---|
| BF16 | 44.22 | 83.18 | 80.07 | 78.61 | 55.29 | 81.82 | 70.53 |
| MXFP4-OCP | 32.50 | 65.37 | 74.41 | 71.19 | 47.61 | 76.43 | 61.25 |
| MX+ (LEE ET AL., 2025) | 34.96 | 69.74 | 74.88 | 71.51 | 47.18 | 77.36 | **62.61** |
| MXFP4-16 | 31.34 | 62.49 | 74.02 | 72.14 | 47.70 | 75.63 | 60.55 |
| MXFP4-16-OAS | 35.02 | 69.50 | 75.99 | 71.27 | 50.68 | 77.90 | 63.39 |
| MXFP4-MBS-S | 37.98 | 74.37 | 77.16 | 73.16 | 52.22 | 79.76 | 65.77 |
| MXFP4-MBS-H | 37.35 | 78.52 | 77.32 | 74.98 | 51.54 | 79.29 | **66.50** |
| NVFP4 | 38.83 | 77.12 | 78.66 | 75.69 | 52.05 | 79.76 | **67.02** |

*Table 4.* Downstream evaluation results on **Qwen3-8B** with different formats.

| PRECISION | MMLU-PRO | GSM8K | HELLASWAG | WINOGRANDE | ARC-C | ARC-E | **AVERAGE** |
|---|---|---|---|---|---|---|---|
| BF16 | 63.14 | 90.38 | 76.51 | 70.56 | 56.91 | 83.33 | 73.47 |
| MXFP4-OCP | 43.78 | 83.83 | 70.98 | 67.01 | 50.77 | 76.64 | 65.50 |
| MX+ (LEE ET AL., 2025) | 51.8 | 86.29 | 72.27 | 68.03 | 49.91 | 77.61 | **67.65** |
| MXFP4-16 | 49.58 | 83.12 | 71.17 | 68.9 | 48.89 | 79.25 | 66.82 |
| MXFP4-16-OAS | 57.85 | 87.52 | 73.14 | 68.03 | 52.56 | 79.17 | 69.71 |
| MXFP4-MBS-S | 58.81 | 87.84 | 73.66 | 68.98 | 52.39 | 81.31 | 70.50 |
| MXFP4-MBS-H | 59.3 | 87.92 | 74.12 | 70.01 | 52.65 | 81.06 | **70.84** |
| NVFP4 | 60.94 | 88.78 | 74.66 | 68.43 | 55.03 | 81.06 | **71.48** |

## 5.2. Benchmarking on Various LLMs

In Table 3 and Table 4, we report downstream evaluation results for different quantization schemes on Llama 3.1-8B-Instruct (L3.1-8B) and Qwen3-8B (Q3-8B). With MXFP4-OCP, L3.1-8B and Q3-8B achieve average accuracies of 61.25% and 65.50% across all benchmarks, respectively. MX+ (Lee et al., 2025), a state-of-the-art MX scheme that repurposes exponent bits in the per-block maximum, improves average accuracy by 1.76% over the MXFP4-OCP baseline. Our MXFP4-16-OAS further improves over MX+ on both L3.1-8B and Q3-8B, with a 1.42% average gain. Building on OAS, applying Macro Block Scaling with the static variant (MBS-S) to both activations and weights yields an additional 1.59% improvement, primarily thanks to better preservation of outliers. Finally, MXFP4-MBS-H (MBS-S for activations and MBS-D for weights) further improves accuracy by 0.54%, reducing the remaining gap to NVFP4 to within 1% on average.

Next, we evaluate our methods on frontier MoE models, including DeepSeek-R1 and Llama 4-Maverick. Consistent with prior observations (Egiazarian et al., 2025), larger models can be less sensitive to quantization; nevertheless, we observe substantial degradation with MXFP4-OCP (up to 10% on MMLU-Pro for DeepSeek-R1; Table 5). Across these models, OAS and MBS substantially recover accuracy, bringing MXFP4 close to (and in some cases on par with) NVFP4. Please refer to Appendix D for Llama

*Table 5.* Downstream evaluation results on DeepSeek-R1 with different formats.

| PRECISION | MMLU-PRO | GSM8K |
|---|---|---|
| BF16 | 83.19 | 95.98 |
| MXFP4-OCP | 72.52 | 95.91 |
| MX+ (LEE ET AL., 2025) | 79.85 | 96.13 |
| MXFP4-16 | 76.29 | 96.66 |
| MXFP4-16-OAS | 75.36 | 96.29 |
| MXFP4-MBS-S | 82.37 | 96.82 |
| MXFP4-MBS-H | 82.06 | 96.89 |
| NVFP4 | 82.69 | 96.36 |

4-Maverick results. We also report perplexity on Wikitext (Merity et al., 2016) in Appendix E, which exhibits the same trend as downstream evaluations and further supports the effectiveness of OAS and MBS. Evaluation on long context benchmark, LongBench (Bai et al., 2024) is available in Appendix F. Additionally, MXFP4-MBS-H can be complementary to PTQ optimizations, and can further improve the results as shown in Appendix G. Overall, these results validate that improved representation fidelity from OAS and MBS translates to consistent end-to-end improvement.

## 5.3. Overhead Analysis

For **MBS-Static (MBS-S)**, we derive $m_{\mathrm{MBS}}^8$ via Equation 3 and scale each $\alpha_i^{16}$ by $(1 + m_{\mathrm{MBS}}^8)$ to obtain the optimized

scaling factor $SF_i$ for the $i$-th sub-block. We develop a CUDA kernel to execute this logic and quantify the instruction overhead using NVIDIA Nsight Compute (NVIDIA Corporation, 2025b). Specifically, while the baseline implementation requires approximately 16.1 ops per element, our Static-MBS approach introduces a marginal average overhead of only 2.7 ops/element. Even with this slight arithmetic increase, the kernel remains strictly bound by data access latency, allowing the additional computation to be effectively hidden behind memory latency. Hence, in our proposed MBS-Static, we observe zero effective overhead for activations which are quantized on-the-fly for MXFP4-MBS-S and MXFP4-MBS-H.

For **MBS-Dynamic (MBS-D)**, we perform an exhaustive search using 16 potential candidates for the $m_{MBS}^8$ that reduces the SSE of the $1 \times 128$ block as shown in Section 4.3.3. In practice, we observe our CUDA implementation to be approximately $2.5\times$–$3\times$ slower than the MBS-S counterpart. We believe it can be further optimized, but for the fair conservative evaluation, we use MBS-H (Hybrid), using MBS-D quantization only for Weights while using MBS-S for activations. This makes sure there is no overhead due to MBS-D during inference as activations still use MBS-S.

Even though the end-to-end evaluations are performed with fake quantization, we also implemented the MXFP4 GEMM kernel with the MXFP4-MBS-H algorithm using MSLK (Meta PyTorch, 2026) and CUTLASS 4.3.0 (NVIDIA Corporation, 2025a) on the NVIDIA Blackwell (SM100) architecture to measure the overhead. The MSLK MXFP4-16 GEMM is implemented with $256 \times 256$ Cooperative Thread Arrays (CTA) tile, and 256 TileShapeK, the cluster shape uses $(2, 4, 1)$. This means each streaming processing (SM) unit will use tensor cores to perform GEMMs at the $256^3$ granularity. The $(2, 4, 1)$ cluster shape means that $2 \times 4 = 8$ SMs will cooperate on a $256 \times 1024$ output tile. As we built on top of this basic implementation, we made a few assumptions in order to be able to reuse this implementation. First, we use 256 for MBS to follow the existing code structure. We compute the E0M8 scaling factor for every 256 elements instead of 128 elements. Second, we assume that there is enough TMEM so we can do triple buffering in the MMA warp so it can overlap with the MBS warps well. Third, in order to make the MBS warp to have similar cost as the MMA warps, we use $128 \times 256$ MBS shape for the weight tensor so we only need one FFMA instruction per element instead of one FFMA and one FMUL instructions per element. Fourth, we assume that there is no warp scheduling contention between the MMA warp and the MBS warps. The detailed implementations are explained in Appendix H. Following these assumptions, the projected overhead using our MXFP4-MBS-H kernel is a 6.2% overhead on top of the baseline GEMM kernel on average for different shapes. For the decode stage, the overhead

is minimal as the execution is memory-bound due to weight loading, consistent with observations in MX+ (Lee et al., 2025). For end-to-end execution time, MXFP4-MBS-H adds negligible overhead for LLM inference.

## 6. Related Work

Prior work explores improving block-based low-precision formats. BDR introduces short microexponents and motivates MX-style formats (Darvish Rouhani et al., 2023). Several methods treat outliers specially, such as by reallocating precision from nearby victim values (Guo et al., 2023), using structured sparsity to mix precisions efficiently (Jeong et al., 2024), or adding interconnect support for heterogeneous bit-widths (Ramachandran et al., 2025). However, most require hardware changes, limiting deployment on commodity GPUs. MX+ is the closest prior work (Lee et al., 2025): it stores extra mantissa bits for the block maximum on top of MXFP4-OCP and runs on GPUs via on-the-fly conversion, but it adds an extra sparse GEMM and can incur up to 54% overhead. Accuracy can be further improved with OAS/MBS-aware calibration (Egiazarian et al., 2025; Frantar et al., 2023; Xiao et al., 2024) or through quantization-aware training, which we leave to future work.

## 7. Conclusion

In this paper, we analyze the MX format and identify the key factors underlying its accuracy gap relative to NVFP4. Based on these insights, we propose OAS and MBS, simple drop-in techniques that strengthen MXFP4. With these enhancements, MXFP4 reduces the accuracy loss of standard MXFP4-OCP by 62% on average, shrinking the gap between MXFP4 and NVFP4 from 10% to <1% on average. Overall, our results show that MXFP4 can deliver near-parity with NVFP4, enabling efficient and accurate quantization for LLM inference.

## Acknowledgements

We thank Preyas Janak Shah for his help with HW overhead estimation. We also thank Shobhit Kanaujia and Chunqiang Tang for their support and guidance throughout the project.

## Impact Statement

This paper presents work whose goal is to advance the field of Machine Learning. There are many potential societal consequences of our work, none which we feel must be specifically highlighted here.

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

## A. Pseudocode for OAS and MBS

OAS modifies E8M0 scale rounding to leverage overflow effectively in the element format as described in the paper. Standard OCP rounding maps blockamax to [4, 8) for E2M1 while clipping for values above 6.0. OAS biases the FP32 mantissa before exponent extraction, shifting the range to (3.5, 7], reducing flush-to-zero for small values while keeping relative error minimal for the largest element. The pseudocode is summarized in Algorithm 1. Note that we return both e8m0_scale and e8m0_unscale in the function. For our fake quantization implementation, we will first multiply the block with the e8m0_scale value so the value is shifted to the E2M1 representation range, apply the E2M1 quantization, up-cast to BF16 type, and then multiply by the e8m0_unscale to map to the original value range.

---

**Algorithm 1** Calculate E8M0 Block Scale Factor

---

**Input:** block_amax (max $|x_i|$ in a block), $n_{\text{mbits}}$ (number of mantissa bits of element format; 1 for E2M1), max_pow2 (largest power-of-two among the representable values for the format; 4 for E2M1)

**Output:** e8m0_scale (power-of-two block scale factor), e8m0_unscale (reciprocal scale factor)

  1: normalized $\leftarrow$ block_amax/max_pow2
  2: oas_bias $\leftarrow (1 \ll (23 - n_{\text{mbits}} - 1)) - 1$
      // For E2M1: oas_bias $= (1 \ll 21) - 1$
      // For E2M1, this causes E8M0 exponent to round up when FP32 mantissa $\geq 0.75$
  3: fp32_bits $\leftarrow$ reinterpret_as_int32($|$normalized$|$)
  4: oased_bits $\leftarrow$ (fp32_bits + oas_bias) & 0x7F800000
  5: exponent $\leftarrow$ (oased_bits $\gg 23$) & 0xFF
  6: e8m0_unscale $\leftarrow 2^{\text{exponent}-127}$
  7: e8m0_scale $\leftarrow 1.0/$e8m0_unscale

---

MBS adds a mantissa-only scale in [1, 2) shared across a macro block of N elements, recovering precision lost by the power-of-two E8M0 scale. The pseudocode is summarized in Algorithm 2.

---

**Algorithm 2** Calculate Macro Block Scale Factors

---

**Input:** $x$ (2D tensor with shape $[M, K]$), mbs_block_size (macro block size, e.g., 128), fp_max (max representable value; 6.0 for E2M1)

**Output:** mbs (scale factors in $[1.0, 2.0)$)

  1: macro_amax $\leftarrow \max(|x|)$ per macro block    // shape: $(M, K/\text{mbs\_block\_size})$
  2: sf $\leftarrow$ fp_max/macro_amax
  3: sf_bits $\leftarrow$ reinterpret_as_int32(sf)
  4: sf_bits $\leftarrow$ (sf_bits & 0x007F8000) $|$ (127 $\ll 23$)
      // This isolates mantissa bits, sets exponent to 0 so result is in $[1.0, 2.0)$
  5: mbs $\leftarrow$ reinterpret_as_float32(sf_bits)

---

## B. Ablation Study with MBS Block Size

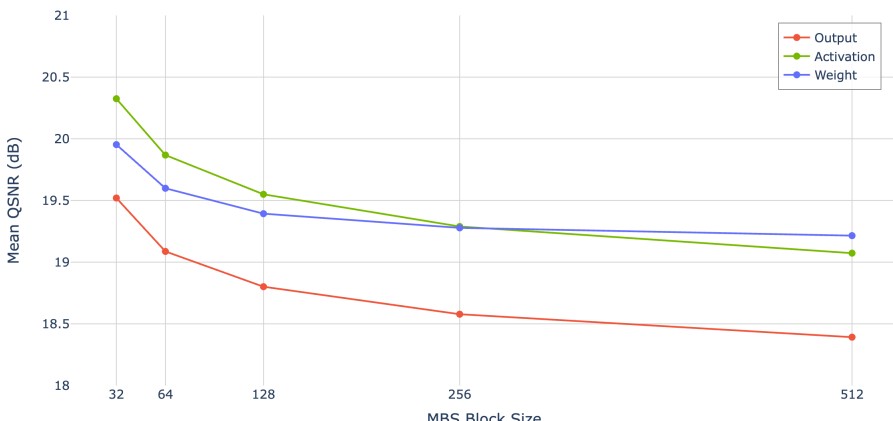

*Figure 7.* Ablation study for MBS block size with Llama 3.1-8B-Instruct.

Figure 7 presents the impact of MBS block size on quantization quality for the Llama3.1-8B-Instruct. We evaluate Mean QSNR across three key components: activation, weight, and output similar to Figure 6.

All three metrics show a consistent downward trend as MBS increases, indicating that larger block sizes lead to degraded quantization quality. The total degradation from MBS=32 to MBS=512 is approximately 1.1 dB for output QSNR, 1.2 dB for activation, and 0.7 dB for weight quantization.

MBS=128 emerges as a favorable operating point, offering a practical balance between quantization quality and hardware efficiency. At this configuration, the model retains 96% of the output QSNR observed at MBS=32.

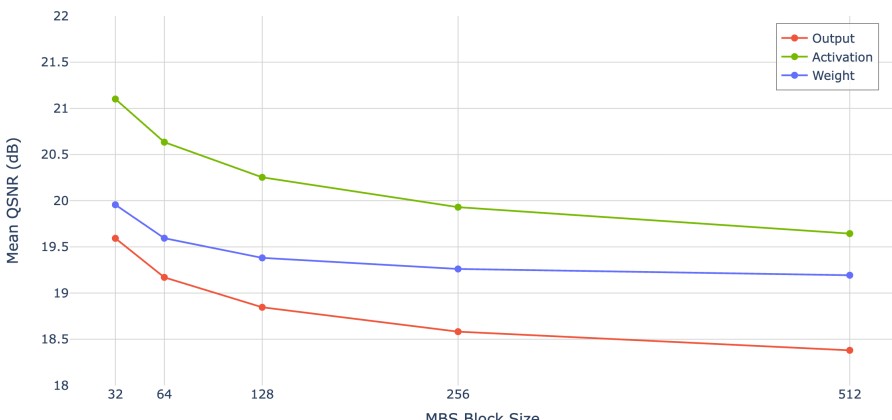

*Figure 8.* Ablation study for MBS block size with Qwen3-8B.

In Figure 8, we show MBS ablation results for the Qwen3-8B. The model exhibits similar degradation patterns to Llama3.1-8B as MBS increases, suggesting consistent quantization behavior across different model architectures. The relatively modest quality degradation of 0.75 dB in output QSNR compared to MBS=32 makes MBS=128 an attractive choice.

## C. Matrix Multiplication Overhead with MBS

**Overhead Analysis**: We analyze the computational and memory overhead introduced by the MBS scaling mechanism, utilizing the target tile configuration of $T_M \times T_N \times T_K = 128 \times 128 \times 128$.

*Computational Overhead:* The baseline FP4 tensor operation performs $2 \cdot 128^3$ FP4 FLOPs per tile ($1 \cdot 128^3$ MAC ops). Our proposed correction adds $2 \cdot 128^2$ FP32 operations (specifically, vector multiplications). The ratio of additional FP32 MULs to baseline FP4 MACs is:

$$\frac{\text{Ops}_{\text{MBS (FP32)}}}{\text{Ops}_{\text{TC (FP4)}}} = \frac{2 \cdot 128^2}{1 \cdot 128^3} = \frac{2}{128} \approx 1.56\% \tag{7}$$

*Memory Traffic Overhead:* For the $128 \times 128$ output tile (64 KB), the MBS scheme loads two scaling vectors ($\sigma_A, \sigma_B$) totaling 512 bytes.

$$\frac{\text{Traffic}_{\text{MBS}}}{\text{Traffic}_{\text{Tile}}} = \frac{512 \text{ bytes}}{65,536 \text{ bytes}} \approx 0.78\% \tag{8}$$

This negligible increase in data movement ensures that the kernel's arithmetic intensity remains mostly unchanged. We also implement the proposed MBS on NVIDIA B200 GPUs and show the analysis in Section 5.3.

## D. Downstream evaluation results on Llama 4-Maverick

*Table 6.* Downstream evaluation results on Llama 4-Maverick with different formats.

| PRECISION | MMLU-PRO | GSM8K |
|---|---|---|
| BF16 | 80.98 | 94.16 |
| MXFP4-OCP | 77.73 | 92.42 |
| MX+ (LEE ET AL., 2025) | 78.46 | 92.80 |
| MXFP4-16 | 78.12 | 92.27 |
| MXFP4-16-OAS | 78.64 | 93.18 |
| MXFP4-MBS-S | 79.38 | 94.16 |
| MXFP4-MBS-H | 79.77 | 93.86 |
| NVFP4 | 80.06 | 94.01 |

In Table 6, we show the evaluation results using different FP4 formats.

## E. Perplexity Evaluation

*Table 7.* Word-level perplexity evaluation on Wikitext with Llama3.1-8B-Instruct and Qwen3-8B using different formats.

| PRECISION | LLAMA3.1-8B-INSTRUCT | QWEN3-8B |
|---|---|---|
| BF16 | 8.82 | 12.20 |
| MXFP4-OCP | 11.49 | 15.18 |
| MX4+ (LEE ET AL., 2025) | 10.82 | 14.51 |
| MXFP4-16 | 11.52 | 15.15 |
| MXFP4-16-OAS | 10.57 | 13.65 |
| MXFP4-MBS-S | 10.04 | 13.09 |
| MXFP4-MBS-H | 9.88 | 13.03 |
| NVFP4 | 9.68 | 12.69 |

In Table 7, we show the perplexity evaluation results using our methods. Using OAS and MBS, we can reduce the perplexity gap against NVFP4 from 1.82 to 0.20 for Llama3.1-8B-Instruct and 2.49 to 0.34 for Qwen3-8B.

## F. Long Context Evaluation

Table 8. Performance comparison across task groups for different formats.

| Task Group | BF16 | MXFP4-OCP | MXFP4-MBS-H (Ours) | NVFP4 |
|---|---|---|---|---|
| Multi-doc QA | 0.435 | 0.319 | 0.392 | 0.388 |
| Single-doc QA | 0.482 | 0.389 | 0.460 | 0.463 |
| Summarization | 0.259 | 0.243 | 0.258 | 0.257 |
| Synthetic | 0.679 | 0.311 | 0.652 | 0.647 |

In Table 8, we show the E2E evaluation scores on the LongBench (Bai et al., 2024) benchmark on the LLaMA-3.1-8B-Instruct model. We observe the same trend as other downstream evaluations: MXFP4-MBS-H significantly outperforms MXFP4-OCP and is comparable to NVFP4.

## G. Applying MXFP4-MBS-H after PTQ

Our primary focus is to improve existing MXFP4-OCP representation fidelity to make it a strong alternative format. OAS/MBS operate at the format level and are complementary to all PTQ strategies, so any PTQ method can leverage OAS/MBS to take advantage of the more accurate MXFP4 representation. To validate this, we conducted an experiment progressively applying each technique, starting with MXFP4-OCP with smaller block size=16 for the fair comparison, then adding SmoothQuant, then OAS rounding mode, and finally MBS:

Table 9. Progressive application of quantization techniques. (Perplexity, lower is better. Relative improvement vs. MXFP4+BS16 baseline.)

| Model | BF16 | MXFP4-16 (baseline) | +SmoothQuant | +SmoothQuant +OAS | +SmoothQuant +OAS+MBS |
|---|---|---|---|---|---|
| Llama-3.1-8B-Instruct | 8.820 | 11.545 | 11.662 (+1.0%) | 10.451 (−9.5%) | 10.010 (−13.3%) |
| Qwen3-8B | 12.199 | 15.155 | 14.681 (−3.1%) | 13.271 (−12.4%) | 13.032 (−14.0%) |

On Qwen3-8B, SmoothQuant alone reduces perplexity from 15.155 to 14.681 (−3.1%). Adding OAS further reduces it to 13.271 (−12.4%), and combining all three techniques (SmoothQuant + OAS + MBS) achieves 13.032 (−14.0%), closing 72% of the gap to BF16 (12.199). On Llama-3.1-8B-Instruct, SmoothQuant alone shows slight regression (+1.0%), but OAS delivers a substantial improvement to 10.451 (−9.5%), and the full combination reaches 10.010 (−13.3%), closing 56% of the gap to BF16 (8.820).

These results demonstrate that OAS and MBS are not redundant with calibration-based PTQ methods, such as SmoothQuant. SmoothQuant redistributes outliers across channels, while OAS optimizes the block scaling factors and MBS recovers mantissa precision lost by the power-of-two E8M0 scale. We believe combining calibration-based PTQ with OAS/MBS more thoroughly is a promising direction.

## H. Detailed Explanation for MBS Implementation

For our MBS implementation, the memory hierarchy spans three levels: global memory (HBM), which stores the FP4 tensors, E8M0 scale factors, E0M8 MBS scales, and final output; shared memory (SMEM), which holds multi-stage buffered tiles for software pipelining; and tensor memory (TMEM), which stores the MMA accumulators and scale factor registers. Data movement adheres to the CUTLASS warp specialization model: producer warp (warp #2) issue asynchronous Tensor Memory Accelerator (TMA) loads for the subsequent K-tile, while consumer warp (warp #0) execute MMA operations on the current tile, synchronized via pipeline barriers. FP4 data and E8M0 scale factors are loaded via TMA with multicast support; subsequently, scale factors are transferred from SMEM to TMEM using Unified Tensor Copy Protocol (UTCCP) operations for block-scaled MMA execution.

The critical algorithmic modification intercepts the MMA inner loop to apply MBS at 256-element boundaries. For each K-tile of 256 elements, the kernel performs block-scaled MMA and writes the per-block partial into a TMEM-

resident accumulator. Directly applying MBS at this stage would block the critical compute path on memory traffic and synchronization. To mitigate this, we adopt a three-stage TMEM accumulator pipeline: at steady state, the MMA warp continually issues the next K-tile's partial into one of three TMEM stages while consumer warps drain a previous stage in parallel, so that the MBS application never blocks the tensor-core issue.

To execute this pipeline efficiently, we extend the stock CUTLASS warp-specialization scheme. The 8-warp baseline assigns warp 0 to MMA, warps 1–3 to scheduling, mainloop load, and epilogue load, and warps 4–7 to epilogue processing; during steady-state mainloop execution the epilogue warps are otherwise idle. We repurpose these idle warps for MBS application and, to further increase SM efficiency for the per-element MBS correction, expand the consumer pool from one warpgroup of four warps to two warpgroups of eight, raising the total warp count from 8 to 12 — warps 0–3 remain the producer pool, and warps 4–11 form the MBS+epilogue consumer pool. The expansion is enabled by per-warpgroup register reconfiguration: producer warps 'setmaxnreg.dec' to a small per-thread budget, while the two consumer warpgroups each 'setmaxnreg.alloc' 232 registers (serialised through a named barrier), leaving enough register pressure for the MBS scale cache and the full-precision accumulator. Within the producer pool we further specialize warp 3: in the stock baseline it is the epilogue-load warp that preloads the $C$ tensor for the $\beta > 0$ case, but for our benchmarking purpose, this work is never executed. We assign warp 3 the new role of MBS scale producer — each K-tile it issues the $MBS_{activation}$ and $MBS_{weight}$ '__ldg' reads, reconstructs the fp32 scaling factors from the compact E0M8 byte representation, folds the two factors into a single per-row scale, and writes the result into a small shared-memory region that the MBS warps subsequently read. Moving this work onto warp 3 frees several SMSP issue cycles per K-tile on every consumer warp that would otherwise compete with the FFMA chain.

Once a TMEM partial is published, each MBS warp copies its assigned fragment from TMEM into registers and fuses the MBS correction directly into its inner FFMA chain—`full_acc + = combined_scale`$[m] \times$ `partial`$[m, n]$, where `combined_scale` is the pre-folded product of the per-row $MBS_{activation}$ and per-block $MBS_{weight}$ produced by warp 3, and `full_acc` is a register-resident accumulator that consumer warps maintain across the entire $K$ dimension. The MBS scales reach the FFMA chain via a single 'ld.shared.v2.f32' per K-tile from a small double-buffered shared-memory region (about 1 KB total across all eight consumer warps and both pipeline stages) that warp 3 writes; the underlying MBS bytes never traverse the TMA path. Two mbarriers coordinate the producer-consumer hand-off: warp 3's per-tile commit becomes a second arrival on the AccumulatorPipeline FULL barrier — alongside the MMA warp's 'umma_arrive_multicast_2x1SM', raising 'producer_arv_count' from one to two — and a complementary 'scales_empty' barrier, arrived once per consumer warp per stage, gives warp 3 backpressure so it never overwrites scales still in use. The MBS correction therefore adds essentially zero TMA or shared-memory-stage overhead on top of the standard MXFP4 mainloop.

