# OpenReview forum: "Unveiling the Potential of Quantization with MXFP4: Strategies for Quantization Error Reduction"
_ICML.cc/2026/Conference — ICML 2026 regular_

### Official Review · Reviewer_k9Xt · 2026-03-03

**Soundness:** 3
**Presentation:** 2
**Significance:** 3
**Originality:** 3
**Overall Recommendation:** 4
**Confidence:** 1

**Summary:**

MXFP4 and NVFP4 exhibit a significant gap in fidelity. The authors trace MXFP4's lower accuracy to its rigid power-of-two scaling factors and coarser block granularity, which struggle to preserve outliers. To mitigate this without requiring hardware modifications, the authors introduce two software-only techniques, Overflow-Aware Scaling (OAS) and Macro Block Scaling (MBS), that improve MXFP4 quantization fidelity without requiring hardware changes. These methods reduce the gap between MXFP4 and NVFP4 to roughly one-tenth of its original size.

**Compliance With Llm Reviewing Policy:**

Affirmed.

**Ethical Review Concerns:**

I am not very familiar with this field. The methods appear to be sound.

**Final Justification:**

I am not very familiar with this field. The methods appear to be sound.

**Key Questions For Authors:**

1. The evaluation appears to omit calibration-based post-training quantization baselines. Modern deployments typically rely on algorithms such as GPTQ, AWQ, or SmoothQuant to handle outliers before format conversion. If weights and activations have already been smoothed, how much additional benefit would the proposed methods provide?
2. Could the authors provide token generation latency benchmark results?
3. Could the authors include a histogram showing the frequency with which the OAS threshold is triggered?

**Limitations:**

See questions.

**Strengths And Weaknesses:**

Strengths:
1. The problem addressed in this paper is important and impactful, and has significant value for the community.
2. The experimental results are substantial: the average accuracy gap is reduced from 10% to 1%.

Weaknesses:
I am not very familiar with this field. The methods appear to be sound. Other weaknesses and limitations can be referenced in the questions below.

---

> ### Author Rebuttal · Authors · 2026-03-31
>
> We thank Reviewer k9Xt for the thoughtful questions and for recognizing the importance and impact of our work. We address each question below.
>
> ----
>
> ### Q1: If weights and activations have already been smoothed, how much additional benefit would the proposed methods provide?
> Our primary focus is to improve existing (OCP) MXFP4 representation fidelity to make it a strong alternative format. OAS/MBS operate at the format level and are complementary to all PTQ strategies, so any PTQ method can leverage OAS/MBS to take advantage of the more accurate MXFP4 representation.
> To validate this, we conducted an experiment progressively applying each technique, starting with MXFP4 (OCP) with smaller block size=16 for the fair comparison, then adding SmoothQuant, then OAS rounding mode, and finally MBS:
> | Model | BF16 | MXFP4+BS16 (baseline) | +SmoothQuant | +SmoothQuant+OAS | +SmoothQuant+OAS+MBS |
> |---|---|---|---|---|---|
> | Llama-3.1-8B-Instruct | 8.820 | 11.545 | 11.662 (+1.0%) | 10.451 (−9.5%) | 10.010 (−13.3%) |
> | Qwen3-8B | 12.199 | 15.155 | 14.681 (−3.1%) | 13.271 (−12.4%) | 13.032 (−14.0%) |
>
> *(Perplexity, lower is better. Relative improvement vs. MXFP4+BS16 baseline.)*
>
> On Qwen3-8B, SmoothQuant alone reduces perplexity from 15.155 to 14.681 (−3.1%). Adding OAS further reduces it to 13.271 (−12.4%), and combining all three techniques (SmoothQuant + OAS + MBS) achieves 13.032 (−14.0%), closing 72% of the gap to BF16 (12.199). On Llama-3.1-8B-Instruct, SmoothQuant alone shows slight regression (+1.0%, which is similar to the observation from [another paper](https://arxiv.org/html/2511.04214v1)), but OAS delivers a substantial improvement to 10.451 (−9.5%), and the full combination reaches 10.010 (−13.3%), closing 56% of the gap to BF16 (8.820).
>
> These results demonstrate that OAS and MBS are not redundant with calibration-based PTQ methods, such as SmoothQuant. SmoothQuant redistributes outliers across channels, while OAS optimizes the block scaling factors and MBS recovers mantissa precision lost by the power-of-two E8M0 scale. While the results above demonstrate clear benefits, we believe combining calibration-based PTQ with OAS/MBS more thoroughly is a promising direction. To ensure fair comparison with prior work (similar to [MX+](https://dl.acm.org/doi/10.1145/3725843.3756118)), we focus on direct-cast quantization in this paper and leave deeper calibration-based PTQ integration as future work.
>
> ### Q2: Could the authors provide token generation latency benchmark results?
> If we measure the GEMM alone, MXFP4-MBS has 6.2% overhead compared to MXFP4-OCP due to the additional MBS scaling factor logic. At the end-to-end level, we integrated our implementation into vLLM and used the it to measure token generation latency (TTIT: time to iterative token). The results are summarized below:
>
> | Model | MX4-OCP TTIT (ms) | MX4-MBS-H (Ours) TTIT (ms) | Overhead |
> |---|---|---|---|
> | Llama-3.1-8B-Instruct | 3.49 | 3.67 | +5.2% |
> | Qwen3-8B | 3.95 | 4.04 | +2.3% |
>
> The overhead on top of TTIT is 2–5%, which includes both GEMM and quantization overhead. We believe this can be further optimized through different techniques, such as fusing quantization with other kernels (e.g., normalization layers), but we leave extensive performance optimization as future work.
>
> ### Q3: Could the authors include a histogram showing the frequency with which the OAS threshold is triggered?
> We analyzed 100 dumped tensors from Llama-3.1-8B-Instruct inference dumps; this includes 23.4M blocks, 1×16 block size, E2M1 with E8M0 scaling.
> The key insight is that the scaling defined by OCP maps the blockmax (amax × local_scale, before E2M1 rounding) to [4, 8), while OAS maps it to (3.5, 7]. Blocks with blockmax in (7, 8) are the ones where OAS triggers, shifting them to (3.5, 4):
>
> | Range | OCP | OAS |
> |---|---|---|
> | (3.5, 4) | 0% | 18.03% |
> | [4, 5) | 32.36% | 32.36% |
> | [5, 6) | 27.07% | 27.07% |
> | [6, 7] | 21.88% | 21.88% |
> | (7, 8) | 18.03% | 0% |
>
> OAS is triggered for **18.03% of all blocks**, shifting blockmax values from the (7, 8) range to (3.5, 4) as they incur larger quantization error. We also found a similar trend on dumped tensors from Qwen3-8B: 18.64% of the blocks were triggered based on the OAS threshold compared to OCP-style scaling.

---

> > ### Author Rebuttal · Reviewer_k9Xt · 2026-04-03
> >
> > Thank you for your reply. My concern has been solved.

---

### Official Review · Reviewer_fneW · 2026-03-04

**Soundness:** 3
**Presentation:** 4
**Significance:** 3
**Originality:** 3
**Overall Recommendation:** 4
**Confidence:** 3

**Summary:**

This paper proposes software-only modifications to MXFP4 to narrow its accuracy gap versus NVFP4. The method combines two independent techniques – Overflow-Aware Scaling (OAS) and Macro-Block Scaling (MBS) – to reduce relative error and improve robustness to outliers. Across multiple LLM families (LLaMA, Qwen, DeepSeek) and benchmarks (MMLU, GSM8K, WinoGrande, etc.), the approach reduces the MXFP4–NVFP4 accuracy gap to < 1%. The authors use QSNR to isolate gains from each technique and report 6.2% overhead over a baseline GEMM kernel, compared to 54% overhead in related work (MX+).

**Compliance With Llm Reviewing Policy:**

Affirmed.

**Key Questions For Authors:**

The current version of the manuscript doesn't contain the link to the repository.  Will the code be made public on the submission?

**Limitations:**

yes

**Strengths And Weaknesses:**

Strengths
-  Shows MXFP4 can match NVFP4 on:
  - Fidelity: within < 1 dB QSNR after applying the
  - Accuracy: within < 1% on average

Weakness:
- The contribution is primarily closing the gap to NVFP4, but doesn't exceed it.

---

> ### Author Rebuttal · Authors · 2026-03-31
>
> We thank Reviewer fneW for the positive assessment and for recognizing the practical value of our work. We would like to clarify that our goal is not to outperform NVFP4 accuracy, but rather to push the limit of existing MXFP4 — which offers significant hardware advantages (simpler E8M0 integer-addition scale resolution vs. E4M3 floating-point multiplication, as detailed in Section 3.4).
>
> ----
>
> ### Q1. Will the code be made public on the submission?
>
> We acknowledge the reviewer's interest in code availability. A core component, specifically one of the main computational kernels central to our approach, has already been implemented, tested, and merged into a popular open-source library. Due to the conference's double-blind review policy, we are unable to disclose the specific library or link at this time, but we are happy to include it in the camera-ready version if accepted. Furthermore, we are actively working through the internal open-source process to release the remaining components publicly after approval.

---

> > ### Author Rebuttal · Reviewer_fneW · 2026-04-01
> >
> > Thank you for your reply - makes all sense.

---

### Official Review · Reviewer_rWw8 · 2026-03-05

**Soundness:** 2
**Presentation:** 2
**Significance:** 3
**Originality:** 3
**Overall Recommendation:** 4
**Confidence:** 3

**Summary:**

The paper explains why MXFP4 typically underperforms NVFP4 for LLM quantization, emphasizing block granularity and power-of-two scaling limits. It proposes two software techniques: OAS (better utilize FP4 range under power-of-two scales) and MBS (apply a coarse higher-precision correction for outliers), and integrates them into GPU GEMM. Experiments across multiple LLMs show MXFP4 can approach NVFP4 accuracy with modest overhead.

**Compliance With Llm Reviewing Policy:**

Affirmed.

**Final Justification:**

I think there are too many important details that are currently only present in the rebuttal, and the paper needs a lot of modification, but overall, my concerns are largely resolved. I therefore maintain my positive score.

**Key Questions For Authors:**

1. What is the actual scale type and block size for the accuracy and throughput evaluations? In Section 4.1, for E8M0 scales, the hardware only supports block size 32. The block size 16 is implemented using E4M3 with wiped mantissa (effectively E4M0). However, in Section 5.3, the MBS kernel is based on MXFP4 GEMM with E8M0 scales and block size 32.

2. If the E8M0 scales are emulated with E4M3, is there any benefit to using the proposed method/kernel instead of directly using NVFP4? Can OAS/MBS methods be generalized to NVFP4 so that the 3 mantissa bits are not wasted and further improve the results?

3. Section 4.4 states "a theoretical equivalence we empirically validate below". What is the theoretical equivalence?

**Limitations:**

yes

**Strengths And Weaknesses:**

**Strengths**

1. The paper gives a clear diagnosis of why MXFP4 loses fidelity relative to NVFP4, which motivates the proposed fixes.

2. The contributions are practical: both OAS and MBS techniques are designed to improve MXFP4 without hardware changes.

3. Results are consistent across several models and benchmarks, with step-by-step improvements that align with the paper's hypotheses. The overhead analysis judges feasibility.

---

**Weaknesses**

1. OAS/MBS expositions in Sections 4.2, 4.3.1, and 4.3.3 are hard to follow, which makes it difficult to verify correctness. Key symbols (e.g., SF, $\alpha$ in Section 4.2, $\mathcal{T}$ in Section 4.3.3) are not defined. The computation steps of the ranges like (3,6] in Section 4.2 are not specified. And the precise steps for OAS/MBS are unclear. It would be helpful to write down the quantization/dequantization equations and/or algorithm pseudocode to show how the exponent and mantissa of the quantized values and scales, the look-up table, and other components in the resulting formats are calculated and used.

2. The area model in Section 3.4 is not fully transparent. The details provided in the paper are insufficient for reproducibility. The claimed tensor core area differences (e.g., 2%, 21.3%, 12.6%) cannot be verified.

3. The GPU overhead analysis in Section 5.3 needs further clarification. The claim of minimal/negligible inference overhead for the decode stage is not supported by measurements. The shape column in Table 4 is ambiguous: what are the batch size, sequence length, and embedding dimensions?

4. Figure 5, the matrix multiplication $C = A \times B$ should be written as $C = A B^\top$.

---

> ### Author Rebuttal · Authors · 2026-03-31
>
> We thank Reviewer rWw8 for recognizing the usefulness of the work, and identifying areas that need more clarity. We address each question below.
>
> ----
>
> ### Q1: What is the actual scale type and block size for the accuracy and throughput evaluations?
>
> We apologize for the confusion. Our research evolved in phases, and we mixed descriptions from different phases, causing inconsistency. To clarify:
> - **Quality studies** (QSNR, E2E evaluation): All results use fake quantization (as in prior work), with **1×16 block size and E8M0 scaling factors** for the base MXFP4 representation, plus additional MBS scaling factors in E0M8 format.
> - **Performance studies**: We initially did not find a native way to perform 1×16 MXFP4 GEMM, so we used NVFP4 with truncated E4M0 to emulate performance. We later discovered that native 1×16 MXFP4 GEMM is achievable in CUTLASS, and we integrated and used that kernel for our performance studies. The emulation approach has been dropped. Please also refer to our response for Reviewer wxpM's Q2.
> - **Section 5.3 ("block size 32")**: This refers to our starting point, the CUTLASS 1×32 block implementation, which we extended to support 1×16 block size and added the MBS feature.
> We will make the description consistent in the camera-ready version.
>
> ### Q2: Can OAS/MBS be generalized to NVFP4 so that the 3 mantissa bits are not wasted?
> As explained above, we use native 1×16 E8M0 scaling in both quality and performance studies, so E4M3 emulation is no longer relevant. Comparing the two formats, E8M0 scaling offers a distinct advantage: its 8-bit exponent provides much wider dynamic range, sufficient to cover any practical tensor without an additional global scale factor. In contrast, NVFP4 uses E4M3 scales whose dynamic range (E2M1 × E4M3) is too narrow, requiring an expensive FP32 per-tensor global scale factor that involves a full-tensor reduction step. Being able to utilize E8M0's wider dynamic range without a global reduction is a key strength of the MXFP4 format as also noted in a concurrent work, [MR-GPTQ](https://arxiv.org/abs/2509.23202) for its higher throughput.
>
> ### Q3: What is the "theoretical equivalence" in Section 4.4?
> We acknowledge the wording is imprecise and will clarify in the camera-ready version. The intended meaning is: we observe a strong positive correlation between QSNR and E2E model accuracy across configurations (as also mentioned in the [previous work](https://dl.acm.org/doi/abs/10.1145/3579371.3589351)). Since OAS+MBS closes the QSNR gap between MXFP4 and NVFP4 to near zero, we can predict, based on this correlation, that E2E accuracy should also converge. We then empirically validate this prediction by showing the E2E accuracy gap is indeed <1% across all benchmarks. The word "theoretical" was used loosely to refer to this QSNR-based prediction; we will revise to "a QSNR-based prediction we empirically validate below" for precision.
>
> ### Others
> - We will also clarify the shape column in Table 4 (M,N,K) in the camera-ready version.
> - Regarding Figure 5: we will fix the matrix multiplication notation. Thank you for catching this.
> - Regarding the reviewer's concern about OAS/MBS exposition clarity, please refer to our response to Reviewer wxpM Q2 and k9Xt Q3.
> - Regarding area model concern, please refer to our response to Reviewer wxpM Q4.
> - Regarding the reviewer's concern about decode-stage overhead, we provide token generation latency, Time-to-Iterative-Token (TTIT) measurements from our vLLM integration. The decode-stage overhead is 2–5%, confirming it is modest.
>
> | Model | MX4-OCP TTIT (ms) | MX4-MBS-H (Ours) TTIT (ms) | Overhead |
> |---|---|---|---|
> | Llama3.1-8B-Instruct | 3.49 | 3.67 | +5.2% |
> | Qwen3-8B | 3.95 | 4.04 | +2.3% |

---

> > ### Author Rebuttal · Reviewer_rWw8 · 2026-04-03
> >
> > Thank you for the response! I will maintain my rating.
> >
> > I think some issues are only partially addressed. The exposition of OAS/MBS remains hard to follow without explicit pseudocode/equations in the paper or rebuttal. The tensor core area model is still not fully reproducible due to the reliance on proprietary implementation details, for instance, the table in the reply to Reviewer wxpM Q4 is still not verifiable.

---

> > > ### Author Response · Authors · 2026-04-08
> > >
> > > We thank the reviewer for the thoughtful follow-up. We address the remaining questions below.
> > >
> > > ### 1. OAS/MBS pseudocode
> > > We provide pseudocode for OAS and MBS below and will include them in the camera-ready version.
> > >
> > > ### 1.1. OAS
> > > OAS modifies E8M0 scale rounding to leverage overflow effectively in the element format as described in the paper. Standard OCP rounding maps blockamax to [4, 8) for E2M1 while clipping for values above 6.0. OAS biases the FP32 mantissa before exponent extraction, shifting the range to (3.5, 7], reducing flush-to-zero for small values while keeping relative error minimal for the largest element.
> > > ```
> > > Input:  block_amax   (max |x_i| in a block)
> > >         n_mbits        (number mantissa bits of element format; 1 for E2M1)
> > >         max_pow2     (largest power-of-two in element format; 4 for E2M1)
> > > Output: e8m0_unscale (power-of-two block scale factor)
> > > 1. normalized <- block_amax / max_pow2
> > > 2. oas_bias <- (1 << (23 - n_mbits - 1)) - 1
> > >    // For E2M1: oas_bias = (1 << 21) - 1
> > >    // This causes E8M0 exponent to round up when FP32 mantissa >= 0.75
> > > 3. fp32_bits   <- reinterpret_as_int32(|normalized|)
> > >    oased_bits <- (fp32_bits + oas_bias) & 0x7F800000
> > >    exponent    <- (oased_bits >> 23) & 0xFF
> > > 4. e8m0_scale <- 2^(exponent - 127)
> > > ```
> > > - Quantization: x_q = round_to_E2M1(x / e8m0_scale)
> > > - Dequantization: x_deq = x_q * e8m0_scale
> > > ### 1.2. MBS
> > > MBS adds a mantissa-only scale in [1, 2) shared across a macro block of N elements, recovering precision lost by the power-of-two E8M0 scale.
> > > ```
> > > Input:  x          (2D tensor with shape [M, K])
> > >         mbs_block_size    (macro block size, e.g., 128)
> > >         fp_max            (max representable value; 6.0 for E2M1)
> > > Output: mbs              (scale factors in [1.0, 2.0))
> > > 1. macro_amax <- max(|x|) per macro block       // shape: (M, K/mbs_block_size)
> > > 2. sf <- fp_max / macro_amax
> > > 3. sf_bits <- reinterpret_as_int32(sf)
> > >    sf_bits <- (sf_bits & 0x007F8000) | (127 << 23)
> > >    // This isolates mantissa bits, set exponent to 0 so result is in [1, 2)
> > >    mbs <- reinterpret_as_float32(sf_bits)
> > > ```
> > > - Quantization: Pre-scale x with x * mbs before MX quantization with OAS.
> > > - Dequantization: Post-unscale x_deq with x_deq / mbs after standard MX dequantization.
> > >
> > > ### 2. Area Model Reproducibility
> > >
> > > We understand that providing reproducible area model is important. During the rebuttal, we also noticed a recent work [INT vs. FP: A Comprehensive Study of Fine-Grained Low-bit Quantization Formats](https://arxiv.org/abs/2510.25602), which provides a publicly available HW cost model that reaches consistent conclusions.
> > >
> > > Their model decomposes a Matrix-Multiply Unit (MMU) into three components, MAC (multiply-accumulate), DEQ (dequantizer / inter-block alignment), and ACC32 (FP32 accumulator). Then, they count standard cells (FA, HA, MUX, AND gates) for each sub-block for the area cost based on TSMC FinFET standard-cell library.
> > >
> > > Using their equations with standard multi-bit adder and multiplier gate counts, we derived the following comparison between MXFP4 and NVFP4 tensor cores using block size 16:
> > >
> > > | Component | MXFP4 | NVFP4 |
> > > |-----------|--------------|--------------|
> > > | MAC       | 4,300        | 4,300        |
> > > | DEQ       | 112          | 1,128        |
> > > | ACC32     | 1,336        | 1,336        |
> > > | Total     | 5,748        | 6,764        |
> > >
> > > The MAC and accumulator costs are identical between formats. The entire overhead comes from the dequantizer (inter-block alignment in our paper's terminology) as we claimed. For MXFP4, dequantization requires only integer additions thanks to E8M0 scales, but NVFP4 dequantization requires floating-point multiplications to resolve the non-power-of-two scale factors (E4M3). Due to the large gap for DEQ, it incurs 17.7% total area overhead, which is consistent with our estimation provided in the table in response to Reviewer wxpM Q4. Furthermore, the table we provided in our response to Reviewer wxpM Q4 covers multiple configurations (block sizes, scale formats), showing the trend holds across different design points.
> > >
> > > We believe this cross-validation with a public methodology, combined with the multi-configuration analysis, addresses the reproducibility question. As the reviewer suggested, we will add this analysis in the camera-ready to strengthen reproducibility.

---

### Official Review · Reviewer_wxpM · 2026-03-11

**Soundness:** 3
**Presentation:** 2
**Significance:** 3
**Originality:** 2
**Overall Recommendation:** 4
**Confidence:** 2

**Summary:**

This paper studies why MXFP4 underperforms NVFP4 and proposes two software-only methods to close the quality gap without hardware changes: Overflow-Aware Scaling (OAS) and Macro Block Scaling (MBS). The paper argues that the main sources of the gap are (i) coarser block granularity and (ii) power-of-two scale precision (E8M0). Reported average accuracy gap to NVFP4 is reduced from ~10% (MXFP4-OCP baseline) to <1%, with modest GEMM overhead.

**Compliance With Llm Reviewing Policy:**

Affirmed.

**Final Justification:**

Thanks for the rebuttal. I keep my positive rating.

**Key Questions For Authors:**

1. Can you provide confidence intervals or multi-run variance for the benchmark averages.
2. Can you release pseudocode and minimal kernels for OAS/MBS to aid reproducibility?
3. How robust is MBS-H under long-context KV-cache-heavy serving and varied batch sizes?
4. Could the reported hardware-area model be accompanied by a public parameterized script?

**Limitations:**

yes

**Strengths And Weaknesses:**

# Strengths
- The problem is important and the paper identifies concrete format-level causes of MXFP4 degradation and ties them to fidelity and hardware-area tradeoffs. To my knowledge is quite interesting.
- OAS and MBS are designed as software-only techniques, which is valuable for deployability.
- The evaluation includes multiple models and tasks and show good results.

# Weaknesses
- Area analysis is analytical and based on a production-like tensor core model, but details are missing, is hard for me to judge if is reasonable. A sensitivity analysis over assumptions (tile size, SRAM/compiler mapping, datapath choices) would improve confidence in the 12.6% area overhead claim for E4M3.
- The paper only reports point estimates but lacks variance across seeds/runs for downstream tasks. For close margins (<1%), confidence intervals matter. And, the statements like “statistically similar errors and comparable inference convergence” are not directly substantiated with formal statistical testing.
-  More explicit end-to-end latency/throughput breakdown by prefill/decode and batch/sequence settings would make the results more clear. Especially if this can be done with different hardware.

---

> ### Author Rebuttal · Authors · 2026-03-31
>
> We thank Reviewer wxpM for the detailed and constructive review. We appreciate the recognition that our problem is important and that OAS/MBS are valuable for deployability. We address each question below.
>
> ----
>
> ### Q1: Variance of results
>
> We ran evaluations using Llama-3.1-8B-Instruct on B200 GPU across WikiText perplexity, GSM8K, and MMLU-Pro, and repeated each evaluation 24 times under identical configurations.
>
> | Benchmark | Std Dev | 95% CI |
> |---|---|---|
> | WikiText (perplexity) | 0.0002 | ±0.0001 |
> | GSM8K (accuracy) | 0.0017 | ±0.0007 |
> | MMLU-Pro (accuracy) | 0.0022 | ±0.0009 |
>
> The variance across all benchmarks is negligibly small (<<1%), far below the accuracy differences reported between configurations in our paper, and does not affect any of our conclusions.
>
> ### Q2: Pseudocode and kernels for OAS/MBS
>
> We provide brief algorithmic descriptions below and will include full pseudocode in the camera-ready version.
>
> - OAS modifies E8M0 scale rounding by adding a bias to the FP32 mantissa bits before exponent extraction: val_to_add = (1 << (23 - m - 1)) - 1, where m is the number of mantissa bits (1 for E2M1). This causes the E8M0 exponent to round up when the FP32 mantissa fraction exceeds the threshold, shifting the blockmax range from [4, 8) to (3.5, 7] and clipping above E2M1's max representable value (6.0).
> - MBS adds a second-level mantissa-only scale factor shared across a macro block of N elements (e.g., 128). It computes sf = fp_max / macro_amax, then extracts the mantissa via bit manipulation ((sf & 0x007FFFFF) | (127 << 23)) to obtain a value in [1.0, 2.0). This pre-scales the data to align the blockmax before MX quantization, recovering precision lost by the power-of-two E8M0 scale.
>
> A core component, specifically one of the main computational kernels central to the effectiveness of our approach, has already been implemented, tested, and merged into a popular open-source library. However, due to the conference's double-blind review policy, we are unable to disclose the specific name of the library or the direct link to the contribution at this time. We are happy to directly link in the camera-ready version of the paper if accepted. Furthermore, we are actively working through the internal open-source process to share remaining components of our work and we intend to release this publicly after the approval.
>
> ### Q3: How robust is MBS-H?
>
> **Long-context evaluation.** We ran LongBench (similar to [TurboQuant](https://iclr.cc/virtual/2026/poster/10006985)) on LLaMA-3.1-8B-Instruct. We observe the same trend as other downstream evaluations: MXFP4-MBS-H significantly outperforms MXFP4-OCP and is comparable to NVFP4.
>
> | Task Group | BF16 | MXFP4-OCP | MXFP4-MBS-H (Ours) | NVFP4 |
> |---|---|---|---|---|
> | Multi-doc QA | 0.435 | 0.319 | 0.392  | 0.388  |
> | Single-doc QA | 0.482 | 0.389  | 0.460 | 0.463 |
> | Summarization | 0.259 | 0.243  | 0.258 | 0.257 |
> | Synthetic | 0.679 | 0.311  | 0.652 | 0.647 |
>
> **Batch size robustness.** We swept batch sizes from 1 to 512 (1, 8, 32, 64, 128, 256, 512) and ran GSM8K evaluation 8 times per batch size. Across all 56 runs, the standard deviation is 0.0067 (<1%), confirming that MBS-H is robust across different batch sizes. This is because MBS is applied along with the hidden dimension, not sequence dimension.
>
> ### Q4: Regarding hardware-area model
>
> Tensor Core (TC) configurations vary across generations and vendors based on (1) M, N, K dimensions (2) spatio-temporal tiling (3) data type throughput choices and (4) design choices such as memory compiler for SRAM, adder tree compression, multiplier algorithm, and rounding mode, which are often proprietary.
>
> Our area analysis is based on component-level area distribution from a production implementation using proprietary EDA flow on an advanced process node, so the absolute numbers cannot be made publicly available. However, the key parameters can be abstracted as two ratios: the **logic fraction** (fraction of tensor core area occupied by compute datapath) and the **multiplier fraction** (fraction of compute datapath occupied by multipliers). This is based on the observations, 1) **Storage area is identical** between NVFP4 and MXFP4. A TC configuration that is more compute-dominated sees larger MXFP4 savings, and 2) **NVFP4 and MXFP4 differ in inter-block alignment logic**; NVFP4 requires expensive FP multipliers. A TC configuration with more expensive multiplier implementations adds to NVFP4 overhead.
>
> | Logic Fraction (at Mul Fraction=0.2) | Relative Area | | Mul Fraction (at Logic Fraction=0.5) | Relative Area |
> |---|---|---|---|---|
> | 0.3 | 0.93 | | 0.10 | 0.94 |
> | 0.4 | 0.91 | | 0.15 | 0.91 |
> | 0.5 | 0.88 | | 0.20 | 0.88 |
> | 0.6 | 0.86 | | 0.25 | 0.86 |
> | 0.7 | 0.84 | | 0.30 | 0.83 |
>
> *Relative area = MXFP4 TC area / NVFP4 TC area*
>
> Across all reasonable parameter choices, MXFP4 achieves 7–17% area reduction over NVFP4 at the tensor core level.

---

> > ### Author Rebuttal · Reviewer_wxpM · 2026-04-03
> >
> > Thanks for the rebuttal. I will keep the positive rating.

---

### Decision · Program_Chairs · 2026-04-30

**Decision:**

Accept (regular)

**Comment:**

**Summary of Paper:**
This paper addresses a practical and timely problem in low-precision LLM inference: the accuracy gap between the Open Compute Project (OCP) MXFP4 format and NVIDIA's NVFP4 format. The authors identify the root causes as MXFP4's coarser block granularity and restricted power-of-two scaling factors. To bridge this gap without hardware modifications, they propose two software-only techniques: Overflow-Aware Scaling (OAS), which optimizes dynamic range utilization, and Macro Block Scaling (MBS), which applies a higher-precision correction at a coarser granularity to preserve outliers. Comprehensive evaluations across multiple LLM families and benchmarks demonstrate that the combined OAS+MBS approach reduces the average end-to-end accuracy gap from ~10% to below 1%, with a modest GEMM overhead of ~6.2%. The work argues this re-establishes MXFP4 as a viable, hardware-efficient alternative to NVFP4.

**Strengths:**

1.	Important and Practical Problem: The work tackles a clear performance-efficiency trade-off (MXFP4 vs. NVFP4) that is highly relevant for real-world LLM deployment.

2.	Clear Technical Contribution: The diagnosis of the format-level deficiencies is insightful, and the proposed OAS and MBS techniques are directly motivated by this analysis. Their software-only nature enhances applicability.

3.	Thorough Empirical Evaluation: The paper provides extensive experiments across multiple models (Llama, Qwen, DeepSeek) and diverse benchmarks (perplexity, MMLU, GSM8K, etc.), convincingly demonstrating the effectiveness of the proposed methods in closing the fidelity gap.

4.	Holistic Analysis: The paper goes beyond accuracy to discuss overhead (GEMM latency, token generation latency) and hardware area advantages, providing a more complete view of the trade-offs.

**Weaknesses and Concerns:**

1.	Initial Clarity and Exposition: Multiple reviewers found the initial description of the OAS and MBS algorithms in the manuscript difficult to follow, with missing definitions and unclear procedural steps. This impacted the ability to verify soundness.

2.	Reproducibility and Statistical Rigor: Initial concerns were raised about the lack of variance estimates for close-margin results and the opacity of the hardware area model, which was based on proprietary data. The presentation of performance results (e.g., batch/sequence dimensions) was also initially ambiguous.

3.	Scope of Contribution: As noted by a reviewer, the core achievement is closing the gap to an existing standard (NVFP4) rather than surpassing it, which somewhat limits the perceived novelty. The idea of MBS also is similar to NVFP4.

**Summary of Rebuttal Process:**

The authors provided a detailed, point-by-point rebuttal that effectively addressed the majority of reviewer concerns. All reviewers acknowledged the improvements.

**Final Assessment:**

The rebuttal successfully strengthened the paper by adding necessary clarity, empirical support, and reproducibility arguments. The core contribution—delivering near-NVFP4 accuracy for MXFP4 through clever software techniques—is technically solid, well-validated, and of clear practical significance to the LLM efficiency community. The remaining minor concerns about exposition are expected to be addressed in the final version. Given the constructive rebuttal and the paper's merits, a Weak Accept recommendation is appropriate. The paper advances the sub-field by providing a compelling software pathway to harness the hardware benefits of the MXFP4 format.